# Understanding the Roles of Very-Long-Chain Polyunsaturated Fatty Acids (VLC-PUFAs) in Eye Health

**DOI:** 10.3390/nu15143096

**Published:** 2023-07-10

**Authors:** Uzoamaka Nwagbo, Paul S. Bernstein

**Affiliations:** 1Department of Pharmacology & Toxicology, University of Utah, Salt Lake City, UT 84132, USA; u.nwagbo@utah.edu; 2Department of Ophthalmology and Visual Sciences, John A. Moran Eye Center, University of Utah School of Medicine, Salt Lake City, UT 84132, USA

**Keywords:** animal models, ELOVL4, lipid dysregulation, very long-chain polyunsaturated fatty acids, VLC-PUFA, retinal diseases, visual health

## Abstract

Lipids serve many roles in the neural system, from synaptic stabilization and signaling to DNA regulation and neuroprotection. They also regulate inflammatory responses, maintain cellular membrane structure, and regulate the homeostatic balance of ions and signaling molecules. An imbalance of lipid subgroups is implicated in the progression of many retinal diseases, such as age-related macular degeneration (AMD), retinitis pigmentosa, and diabetic retinopathy, and diet can play a key role in influencing these diseases’ onset, progression, and severity. A special class of lipids termed very-long-chain polyunsaturated fatty acids (VLC-PUFAs) is found exclusively in mammalian vertebrate retinas and a few other tissues. They comprise <2% of fatty acids in the retina and are depleted in the retinas of patients with diseases like diabetic retinopathy and AMD. However, the implications of the reduction in VLC-PUFA levels are poorly understood. Dietary supplementation studies and *ELOVL4* transgene studies have had positive outcomes. However, much remains to be understood about their role in retinal health and the potential for targeted therapies against retinal disease.

## 1. Introduction

Very-long-chain polyunsaturated fatty acids (VLC-PUFAs) were first isolated and analyzed in bovine retinas by Aveldaño in 1987 [1]. They are now classified as fatty acids with greater than 24 carbons on the longest continuous carbon chain. They have a hybrid structure combining a proximal carboxylic end characteristic of saturated fatty acids and a distal end containing methylene-interrupted cis double bonds [2] (Figure 1). Most retinal VLC-PUFAs are synthesized in situ and de novo by ELOVL4 and other biosynthetic enzymes through tandem elongation and desaturation steps (Figure 2). Retinal VLC-PUFA species predominantly belong to the n-3 and n-6 families [3,4], and VLC-PUFAs with up to 13 double bonds have been identified in vertebrate tissue [5]. VLC-PUFAs are relatively low in abundance and found exclusively in highly specialized tissues such as the retina, brain, meibomian gland, and gonads of mammals and other vertebrates [6]. Aside from animal tissue, VLC-PUFAs can be obtained by extracting and purifying them from cells overexpressing ELOVL4 or synthetically from organozinc coupling reactions between a long-chain polyunsaturated fatty acid (LC-PUFA) and saturated fatty acid [7].

VLC-PUFAs are important in maintaining photoreceptor structure, function, and health. Research into the mechanisms of age-related macular degeneration (AMD) pathology shows lipid dysregulation is an underlying factor in AMD and other retinal dystrophies. Patients with AMD were found to have lower levels of VLC-PUFAs and omega-3 to omega-6 (n-3/n-6) fatty acids ratios compared to healthy controls. The low abundance and difficulty synthesizing VLC-PUFAs have made them difficult to study until recently. Moreover, their confinement to specialized tissue makes them not easily obtainable through a Western diet. Thus, assessing how these lipids affect eye health has been difficult, especially in patients with blinding diseases like diabetic retinopathy and AMD. In this review, we will discuss the current knowledge of VLC-PUFAs concerning eye health and disease and the strategies used to study their function in vivo and in vitro.

## 2. Importance of VLC-PUFAs to Eye Health

While much research exists on the relevance of LC-PUFAs, such as docosahexaenoic acid (DHA), to infant development and eye health, epidemiological studies on VLC-PUFAs’ relevance to eye health and development are scarce. Dietary supplementation with omega-3 fatty acids such as DHA has beneficial effects on membrane stability, calcium signaling, and as precursors for anti-inflammatory molecules. High intake of omega-3-rich diets seems to have neuroprotective effects against retinal degeneration in patients with AMD [11], Stargardt-3 disease (STGD3) [12], diabetic retinopathy [13], and retinitis pigmentosa [14]. DHA is enriched in photoreceptors and makes up 40–60% of the fatty acids in the retina. DHA also serves as a precursor for anti-inflammatory docosanoids and neuroprotectins, maintains membrane fluidity, and interacts with rhodopsin to facilitate phototransduction.

The retina has one of the highest contents of PUFAs in all human tissues. As the retina is regularly exposed to oxidizing light damage, retinal lipids—including beneficial LC-PUFAs, such as DHA, are susceptible to lipid peroxidation. Aging retinas are even more vulnerable to lipid peroxidation due to the accumulation of A2E and iso-A2E, which are pyridinium bisretinoids that contribute to drusen formation in dry AMD [15]. Concentrations of LC-PUFAs and VLC-PUFAs in the retina, retinal pigment epithelium (RPE), and choroid decrease with age and more substantially in patients with diseases such as AMD and diabetic retinopathy [8,16]. Maintaining a healthy balance of unsaturated to saturated fatty acids is important to eye health because saturated fatty acids accumulate in retinal tissue with age [17].

## 3. ELOVL4 Structure and Biosynthesis of VLC-PUFA

VLC-PUFAs are biosynthesized in cells expressing the enzyme, elongation of very-long-chain fatty acids-4 (ELOVL4). ELOVL4 is a 3-keto acyl-CoA synthase that catalyzes the rate-limiting condensation step of very long-chain fatty acid (VLC-FA) synthesis from shorter-chain fatty acid precursors. After condensation by an ELOVL enzyme to attach two-carbon units to the fatty acyl chain, the fatty acid goes through three other enzymatic steps within the elongase complex with associated enzymes, including reduction with a β-ketoacyl-CoA reductase; dehydration with a β-hydroxy acyl-CoA dehydratase; and another reduction with an enoyl-CoA reductase. Through a series of elongation and desaturation reactions, VLC-PUFAs of different chain lengths and degrees of unsaturation are synthesized from dietary precursors. The elongated fatty acyl-CoA is metabolized by the cell or used as a substrate for further elongation [18]. Like other ELO-protein family members, ELOVL4 is directed to the endoplasmic reticulum (ER) through its conserved di-lysine KXKXX ER-retention motif. Meanwhile, its histidine-rich HXXHH binding motif is used for electron transfer and redox reactions during fatty acid elongation [19]. ELOVL4 can be found in the inner segments of rod and cone cells in the retina [10], and its gene has one splice variant and six exons [20]. In humans, its protein has 314 amino acids, is 36.8 kDa in size, has seven transmembrane domains, and has no signal peptide [21].

Three mutations in the ELOVL4 gene are known to cause STGD3, a rare form of autosomal dominant early-onset macular degeneration and an orphan disease in the United States. These mutations are found in Exon 6 and include a 5-bp deletion (rs1131690770), two 1-bp deletions (rs587776613), and a C > G transversion mutation (rs104893946), all of which lead to a truncated protein product which mislocalizes from the ER. Two other recessive variants in trans in the promoter region of ELOVL4 were found in one Stargardt patient; the authors provide evidence that these variants could lead to the downregulation of ELOVL4 expression [22]. More recently, another variant in Exon 1 of ELOVL4, c.59A > G (p.Asn20Ser), was found in a Chinese Stargardt patient [23]. While ELOVL4 dysfunction is the basis of STGD3 pathology, it is unclear how the mislocalized truncated ELOVL4 protein, the protein’s loss of catalytic activity, or a combination of these factors contributes to the pathogenesis and progression of the disease. Understanding the role of the loss of VLC-PUFAs and enzymatic activity of ELOVL4 is pertinent to understanding the etiology of the more common neurodegenerative disease, AMD, whose patients share similar retinal depletions of VLC-PUFAs (Figure 3) [16].

Other mutations in ELOVL4 have been associated with spinocerebellar ataxia, dry eye, ichthyosis, mental retardation, nystagmus, poor eye contact, latent visual responses, reduced electroretinogram (ERG) responses, retinitis pigmentosa, and myopathy [24,25,26,27,28,29,30]. Some of these ocular manifestations may be attributed to VLC-PUFA loss from mutant ELOVL4. However, interestingly, Agbaga et al. [28] found that the W246G mutation also in Exon 6 of Elovl4 selectively impaired Elovl4’s ability to synthesize very long-chain saturated fatty acids (VLC-SFAs) but not VLC-PUFAs in the retina and skin of rats. They suggested that although these rats did not present any evidence of retinal degeneration, there was a possibility of neural synaptic transmission impairment in the retina. Moreover, these rats also present with swollen eyelids and localized hair loss around the eyes, nose, and ears [28,31]. In contrast, Exon 6 mutations in Elovl4 associated with STGD3 impaired both VLC-PUFA and VLC-SFA synthesis [32]. This suggests that a change in the fatty acyl binding site or conformational state of the Elovl4 protein specifically allows the binding or docking of saturated or polyunsaturated fatty acids and that this is impaired with the W246G and STGD3 mutations in Exon 6. A mutational analysis of the ELOVL4 protein would be needed to confirm this. Interestingly, a structural analysis of the ELOVL7 protein by Nie et al. [33] confirmed a binding pocket for saturated and unsaturated fatty acyl chains that included residues from the protein’s final transmembrane domain.

## 4. VLC-PUFAs

### 4.1. Localization of VLC-PUFAs in the Eye

VLC-PUFAs make up <2% of total fatty acids in the retina. There are enriched in the photoreceptor outer segment disks of the retina. VLC-PUFAs in the eye are commonly found in the sn-1 position of di-polyunsaturated phosphatidylcholines, with DHA attached to the sn-2 position [4]. The metabolic processes governing this chemical arrangement are unknown. VLC-PUFAs in other tissues have been documented in ceramide, sphingomyelin, cholesteryl ester, phosphatidylcholine, and triacylglycerol lipid forms [34,35]. For instance, they can be found in sphingomyelin and ceramides in rodent spermatozoa [36] and skin [37]. Their detection limit, length, abundance, and degrees of unsaturation depend on their location and assay conditions. The composition of lipids and VLC-PUFAs in rods, cones, and other retinal cells also differs [38,39,40]. For instance, cone outer segments (COS) have less DHA and phosphatidylethanolamine than rod outer segments (ROS). They also differ in the plasma membrane composition of VLC-PUFAs of specific chain lengths [38,39]. These differences in the VLC-PUFA profile may explain cone-specific dystrophy in macular degenerations.

### 4.2. Physiological Roles of VLC-PUFA

In contrast to diseases of ELOVL4 dysfunction and VLC-PUFA deficiency, VLC-PUFA overabundance due to peroxisomal dysfunction is known to cause several diseases under the Zellweger Spectrum of Disorders (ZSD). Peroxisomes are responsible for very-long fatty acid β-oxidation and degradation. ZSD causes craniofacial abnormalities, reduced DHA levels, and increased levels of VLC-PUFAs in plasma and various tissues [41]. ZSD peroxisomal dysfunctions mostly occur through PEX or ADCD1 transporter gene mutations. Hearing loss and vision disorders like hypertelorism, cataracts, nystagmus, and retinal dystrophies are seen in children with ZSD [42,43]. Moreover, retinal degeneration has been linked to the dysfunction in peroxisomal protein, multifunctional protein 2 (MFP2) [44]. Thus, both VLC-PUFA deficiency and overabundance can negatively impact visual health.

### 4.3. VLC-PUFAs in the RPE

The retinal pigment epithelium (RPE) is a monolayer of pigmented polarized cells whose apical side faces the photoreceptor outer segments and whose basal side faces the choroid. Aside from nutrient and metabolic support, the RPE offers structural support to photoreceptors and phagocytoses rod and cone outer segments, which is crucial for recycling photopigments. VLC-PUFAs have been implicated in RPE health in various studies, and the RPE phagocytoses VLC-PUFA-enriched photoreceptor outer segments [39]. STGD3 pathology also causes cytotoxicity in RPE cells and reactivity in microglia and macrophages [45]. Evidence of ELOVL4 activity in the RPE is disputable. Esteve-Rudd et al. suggested that since ELOVL4 is not expressed in the RPE, STGD3 pathology in the RPE must be due to the effect of the RPE cells phagocytosing photoreceptor cells with mutant ELOVL4 protein [46]. Nevertheless, ELOVL2 (upstream of ELOVL4 in the VLC-PUFA biosynthetic pathway) has been reported to be expressed in RPE cells [47]. Moreover, the shRNA knock-down of ELOVL2 results in cell senescence and other impairments in primary human RPE cells [48].

Omega-3-rich diets are also known to improve RPE health. For instance, mice with photoreceptor-specific activated MTORC1 (to recapitulate AMD pathology) were fed DHA and were reported to have enhanced phagocytosis of photoreceptor outer segments by the RPE and a reduction of the other retinal symptoms associated with AMD [49]. More research needs to be done to assess if VLC-PUFA supplementation would have a similar effect on RPE health in humans and rodent models of AMD.

### 4.4. VLC-PUFAs in Neurotransmission

VLC-PUFAs have been linked to neurotransduction in photoreceptors. In addition to being situated in photoreceptor outer segments, they are also located in photoreceptor ribbon synapses [50]. Very long-chain saturated fatty acids (VLC-SFAs), which are more commonly found in the skin and brain, regulate the exosomal release of synaptic vesicles in neuronal cells [10]. However, the role of ELOVL4-mediated synaptic transmission in the retina and whether VLC-PUFAs also perform this role still needs to be determined, even though ELOVL4 biosynthesizes both VLC-FAs.

### 4.5. Biophysical Chemistry

VLC-PUFAs and ELOVL4 have been associated with rhodopsin, the main G-protein coupled receptor (GPCR) involved in phototransduction [3,21,51]. Data show that due to their high degree of unsaturation and length, the VLC-PUFA products of ELOVL4 can uniquely impact membrane bilayers. Cheng et al. found that the C32:6 n-3 VLC-PUFA (enriched in rod outer segments) may promote lipid translocation across photoreceptor outer segment bilayers. Lipid translocation helps translocate retinoids (the main component of visual chromophores) across a membrane bilayer [52].

Although their role in cellular processes remains unclear, their unique hybrid structure has led researchers to believe VLC-PUFAs maintain the curvature of photoreceptor disks by their unique ability to occupy both leaflets of a phospholipid bilayer. Their concentration and orientation in the plasma membrane may also allow them to uniquely interact with and differentially stabilize certain proteins and molecules in photoreceptor disks’ inner and outer laminae [2]. VLC-PUFAs alter membrane dynamics. The additions of a physiologically relevant mole percentage of VLC-PUFA (C32:6 n-3, 0.1 mol %) to a distearoylphosphatidylcholine (DSPC) bilayer greatly increased lipid flip-flop rate (Figure 4) and bilayer compression modulus by increasing surface pressure between 10 mN/m and 50 mN/m. The same effect was not observed with DHA or higher VLC-PUFA concentrations in the membrane [52]. Retinal alterations of VLC-PUFA have also been linked to A2E accumulation and photoreceptor death in mouse models of STGD3 [53]. While the mechanism of this association is unclear, there is speculation that VLC-PUFAs may be involved in non-enzymatic retinoid translocation in the retina. There is also some speculation that VLC-PUFAs may play a non-enzymatic role in the intraluminal translocation of *n*-retinylidene phosphatidylethanolamine (*n*-retinyl-PE), which is why STGD3 pathology somewhat phenocopies ABCA4-related pathology in STGD1 disease [54].

### 4.6. ELOVL2 and VLC-PUFAs

ELOVL2 is one of the elongases upstream of ELOVL4 on the VLC-PUFA biosynthetic pathway. It is another member of the seven-member family of ELOVL elongases responsible for LC-PUFA elongation in humans. ELOVL2 has also been implicated in AMD and retinal health, and mutations in this gene have been used to model visual abnormalities in mice, zebrafish, and cell culture models [55,56], lending credence to the importance of VLC-PUFAs in retinal health. Photoreceptor loss is also associated with reduced DHA and Elovl2 expression in the retinas of AdipoR1 deficient mice, as AdipoR1 is thought to induce ELOVL2 expression [55]. DNA methylation of ELOVL2’s promoter region is also an epigenetic marker for aging, and methylation reversal with 5-aza-2′-deoxycytidine (5-aza-dc) restores Elovl2 expression and vision in mice [47]. Elovl2-deficient zebrafish had diminished visual motor function and reduced levels of C24 VLC-PUFA substrates [56]. It is unclear if ELOVL4 has epigenetic modifications which can be reversed to restore a normal phenotype.

### 4.7. VLC-PUFA Derivatives

Omega-3 and omega-6 PUFAs are critical for membrane stability and cell signaling within the retina. They are also precursor molecules for important inflammatory regulators such as prostaglandins and leukotrienes. Although it is unclear if VLC-PUFAs serve as signaling molecules or immune modulators, there is research on elovanoids, which are C32 (ELV-N32) or C34 (ELV-N34) n-3 VLC-PUFA-derivatives involved in many signaling and metabolic processes. Elovanoids are lipid mediators that regulate neuroprotection, immune response, and apoptosis in the retina and central nervous system. Elovanoids are not the only PUFA-derived neuroprotectants, as DHA-derived docosanoids and EPA-derived eicosanoids are also involved in similar activities against neurodegeneration [57]. These neuroprotectants are generated under cellular stress conditions where the fatty acid is liberated from the phospholipid and converted to the neuroprotectant [58]. VLC-PUFAs have also been found as C32-C36 O-acyl-ω-hydroxy fatty acid (OAHFA) trienes in human meibum secretions [59]; however, a biosynthetic pathway of these lipids from VLC-PUFAs has not been confirmed.

## 5. Dietary Studies

The effect of DHA supplementation on visual function has been well-studied; however, only a few studies exist on the effect of VLC-PUFA supplementation on vision. Oral gavage studies show that C32:6 n-3 VLC-PUFA is bioavailable in mice’s eyes and retinas and improves visual function [34,60]. Similarly, transgenic overexpression of ELOVL4 improved visual outcomes in a mouse model of diabetic retinopathy [61]. In addition, it has been shown that ELOVL4 promotes the differentiation and healthy accumulation of lipids in neuroblastoma cells but is transcriptionally regulated by the MYCN oncogene, which has severe implications on disease outcomes for neuroblastoma patients [62].

Recently, mice fed deuterated docosahexaenoic acid (D-DHA) also exhibited increased D-DHA levels in the retina, reduced oxidative damage, and improved visual function compared to mice fed chow supplemented with non-deuterated DHA [63,64]. Deuterium was also found to be incorporated into VLC-PUFAs, consistent with the literature on DHA as a VLC-PUFA precursor. Dietary studies involving deuterium-labeled VLC-PUFAs may prove to be a good way to accurately determine the effective dose, bioavailability, and stability of VLC-PUFAs in animal models of retinal degeneration. However, obtaining commercially available deuterated or chemically-labeled VLC-PUFAs for large-scale dietary studies in humans and animals is challenging.

Furthermore, as an indirect route, research has shown that fatty acids modify gut microbiota and influence γ-aminobutyric acid (GABA) signaling in the gut-brain axis [65]. The added signaling effect would affect the therapeutic potential of oral compared to intravenous administration routes. Studies have also linked the gut microbiome with retina and lens lipid composition [66], and some of the changes in gut microbe composition and their effects on the brain and liver lipidomes are age-dependent [67]. Interestingly, diet and culture conditions can also regulate Elovl4 expression and tissue-specific VLC-PUFA distribution in fish [68]. ELOVL4 expression also changes temporally and in disease conditions [10,69], thus affecting VLC-PUFA composition and abundance in different tissues at any given time.

## 6. Cell Culture Studies and Substrate Specificity

Although ELOVL4 is known to synthesize VLC-PUFAs >24 carbon in chain length, little is known about its tissue-specific substrate specificities for fatty acids of different chain lengths and degrees of unsaturation. Cell culture studies have revealed ELOVL4’s substrate specificity through fatty acid elongation studies in primary and transgenic cell lines [2]. ELOVL4 preferentially synthesized omega-3 VLC-PUFAs from EPA over DHA in transgenic and primary cell cultures [19,70,71]. However, the preferential elongation of specific fatty acids by ELOVL4 is known to depend on substrate concentration and tissue [72].

Like mammals, Elovl4 is highly expressed in the eyes, brain, and gonads of zebrafish and also synthesizes many of the same fatty acids as human ELOVL4 [9]. Fish VLC-PUFAs are also mostly found in phosphatidylcholines and have tissue-specific enrichment of VLC-FAs of different chain lengths and degrees of unsaturation [35]. Moreover, unlike humans, through a teleost-specific (3R) whole genome duplication event, zebrafish possess two Elolv4 paralogues, *Elovl4a*, and *Elovl4b*, with different spatiotemporal expression patterns, substrate specificities, and sub-functionalizations [9]. While zebrafish are unlikely to dehydrate from the deletion of Elovl4 due to their aqueous habitat, their duplicate *Elovl4* genes, *Elovl4a,* and *Elovl4b,* have distinct substrate specificities for saturated compared to unsaturated VLC-FAs when analyzed in yeast tested through heterologous gene expression. While Elovl4a primarily elongated VLC-SFAs, Elovl4b was the primary enzyme responsible for VLC-PUFA synthesis [9]. Nonetheless, it is still important to note that since ELOVL4 expression and VLC-PUFA distribution can be extrinsically regulated [68], ELOVL4 substrate specificities in human tissues may be vastly different from those observed in cell culture and other organisms.

## 7. Animal Models

Animal and cell culture models with dysfunctional ELOVL4 proteins can serve as tools to study VLC-FA depletion. Since no specific pharmacological inhibitors of ELOVL4 or VLC-PUFA biosynthesis exist, scientists have mostly relied on genetic models to study the role of ELOVL4 and VLC-PUFAs in health and disease. Mice with *Elovl4* knock-in and knockout mutations have been generated to understand the mechanism of STGD3 pathology in the retina. These studies have presented conflicting results in mutant mice of varying phenotypes because STGD3 is a complex disease involving many cell types, and it is unclear what role *ELOVL4* haploinsufficiency and VLC-PUFA-depletion play in STGD3 pathology.

Global knockout of *Elovl4* is lethal in postnatal mice, as they dehydrate shortly after birth due to the loss of ceramides in their skin [10]. This is also the case for mice homozygous for STGD3 mutations [32]. Some studies report that transgenic mice expressing varying copy numbers of the human (5-bp del) STGD3 gene (under the photoreceptor-specific Interphotoreceptor Retinoid-Binding Protein “IRBP” promoter) have progressive photoreceptor degeneration and other symptoms reminiscent of human pathology [73]. In this model, the severity of the disease was linked to the copy number of the STGD3 transgene [10]. This finding was similar to the STGD3 knock-in mutant by Vasireddy et al. [74], wherein rod and cone a- and b-wave responses were enhanced, but photoreceptor degeneration was observed. Conversely, McMahon et al. reported that their heterozygous 5-bp deletion STGD3 knock-in mice have normal retinal morphology but reduced ERG a- and b-waves and increased lipofuscin accumulation. Thus, the STGD3 mutants display varying phenotypes depending on the mutagenesis strategy, induction times, and cellular targets.

A developmental origin of STGD3 pathology was challenged in studies by Li et al., where homozygous STGD3 mouse neonates appeared to have normal retinal morphology [32]. Although retinal morphology might seem normal, it is unclear if these neonates had abnormal ERGs, lipid deposition in their retina or RPE, other visual behavioral responses, or if healthy maternal *Elovl4* mRNA still influenced the phenotype at the neonatal stage.

Other sources have claimed that haploinsufficiency is not the cause of STGD3 pathology, as *Elovl4*-knockout mice had normal retinal development before death, and heterozygous knockout mice even had superior a- and b-wave photoreceptor responses on an electroretinogram [75]. Two transgenic pigs modeling the 5-bp deletion STGD3 mutation had reduced ERG a- and b-waves, mislocalized mutation protein, and retinal layer thinning and disorganization [76].

Moreover, Barabas et al. compared rod-cone conditional *Elovl4* conditional knockout mice to Karan et al.’s STGD3 transgenic mice. They found that the tissue-specific conditional knockout of the *Elovl4* did not recapitulate STGD3 pathology. However, the caveat is that the tissue-specific knockout mutations did not completely deplete VLC-PUFA levels in adult mouse retinas [77], which may indicate that retinal *Elovl4* expression is not limited to cells with the rod-specific opsin and cone-specific human red-green pigment (HRGP) promoters used to generate the conditional knockouts. More research is needed on if mutant and wild-type *ELOVL4* or its enzymatic products can diffuse or be transported into other cells surrounding the photoreceptors.

Copy number variations (CNV) are another source of variability in the STGD3 mouse models. CNVs have been implicated in the etiology of many inherited retinal dystrophies (IRDs) [78]. Cell lines overexpressing *ELOVL4* showed significant abnormalities and signs of cellular stress. In contrast, animal studies by McMahon et al. found no signs of cellular stress in their heterozygous STGD3 mutant mice, lending credence to the school of thought that VLC-PUFA depletion rather than cellular stress underlies STGD3 pathology [79]. A detailed comparison of the mouse models of STGD3 can be further explored in the 2019 paper by Hopiavuori et al. [10].

## 8. VLC-PUFA Analysis Tools

The last review of the methods and tools used in VLC-PUFA analysis was done by Berdeaux and Acar in 2011 [80]. Since then, new technologies have emerged, and novel VLC-PUFA metabolites and lipid groups containing VLC-PUFAs have been identified. This section will discuss the old and new methods used to analyze these groups.

The first steps of VLC-PUFA analyses usually involve total lipid extraction from homogenized tissue or cell cultures through supercritical fluid extraction or solvent-based methods such as Folch, Bligh and Dyer, or Matyash. These methods prevent oxidative damage or thermal degradation of fatty acid analytes. Since VLC-PUFAs are quite low in abundance, samples from small animals, cell cultures, or organoids are typically pooled and concentrated to obtain a detectable signal. Subsequently, VLC-PUFA-containing lipid groups can be separated through thin layer chromatography (TLC), derivatized into fatty acid methyl esters (FAMEs), concentrated, then resuspended in a suitable analysis buffer. The concentrated analyte is then subjected to gas chromatography-mass spectrometry (GC-MS) or liquid chromatography-mass spectrometry (LC-MS). And finally, post-analysis peak integration and lipid quantification. LC-MS/MS is a good way to understand the complex lipid groups containing VLC-PUFAs with FAME derivatization. At the same time, GC-MS is a great way to analyze VLC-PUFA concentrates in low quantities. Some groups have skipped the chromatographic step entirely and directly analyzed VLC-PUFAs in complex lipids using ionization and MS strategies [40,79,81].

Different forms of ionization exist to generate precursor and fragment ions for mass-spectrometry analysis. Hard ionization leads to extensive fragmentation and better compound identification because mass spectrometry detects each molecule’s unique fragmentation pattern. On the other hand, soft ionization ionizes the lipids but retains the precursor mass-to-charge ratio (*m*/*z*) as much as possible to identify larger, more complex compounds.

### 8.1. GC-MS

One of the main ways to analyze VLC-PUFAs is gas chromatography-mass spectrometry (GC-MS). For GC-MS analysis, total fatty acids are extracted and converted to FAMEs before solid phase extraction or reverse phase TLC, concentration, and then GC-MS analysis [4]. Aveldaño’s group analyzed VLC-PUFAs in bovine retinas through a series of separation steps using TLC, argentation thin layer chromatography (AgNO_3_-TLC), and high-performance liquid chromatography (HPLC) before oxidative ozonolysis and gas chromatography-mass spectrometry with electron impact ionization (GC-EI-MS) analyses. Through these methods, they could identify VLC-PUFAs by their degree of unsaturation, chain length, and lipid species. They identified both omega-3 and omega-6 penta-unsaturated VLC-PUFAs, omega-3 hexa-unsaturated, and omega-3 tetra-unsaturated VLC-PUFAs in bovine retinas by oxidative ozonolysis [1,3].

Subsequently, in 1994, Suh et al. [82] extracted and analyzed VLC-PUFAs from normal and diabetic rat rod outer segments. They first isolated total lipids and then converted VLC-PUFA-containing fractions to FAMEs by hydrolysis, followed by AgNO_3_-TLC to separate FAMEs based on the degree of saturation. After the removal of cholesterol and saturated fatty acid impurities, they then injected FAMEs of each band for GC-EI-MS analysis and identified positive precursor and fragment ions [M+} on their MS analyses. Suh et al. determined that *m*/*z* 79 was the diagnostic fragmentation ion for PUFAs. They also determined that *m*/*z* 108 fragment ions identified *n*-3 PUFAs, while *m*/*z* 150 fragmentation ions are from the *n*-6 series [82]. The ratios of the ions have since been used to quantify *n*-3 and *n*-6 VLC-PUFAs. In 2008, Agbaga et al. modified Suh et al.’s method to identify low-concentration VLC-PUFAs in cell culture through selective ion monitoring (SIM) mode mass spectrometry [19,80].

In 2010, Liu et al. developed a method to quantify VLC-PUFAs without using TLC to purify FAMEs. They performed solid-phase extraction using silica gel cartridges to remove cholesterol contaminants and reduce processing time to avoid lipid peroxidation. They used a combination of liquid chemical ionization (LCI) and electron ionization to identify *n*-3 and *n*-6 VLC-PUFAs, then SIM to quantify VLC-PUFAs based on fragmentation ions with *m*/*z* 79, 108, and 150 as used by Suh and Agbaga [16], and matching lipid extract peaks’ retention times to bovine retina VLC-PUFA retention times [19]. This method has been used to assess VLC-PUFAs in human and diet-supplemented rodent retinas [60,83,84].

Serrano et al. [85] recently quantified VLC-PUFAs in the fish *Sparus aurata* using positive atmospheric pressure chemical ionization (APCI). This soft ionization mode was coupled with gas chromatography and a quadrupole orthogonal acceleration time-of-flight mass spectrometer source. Lipids were extracted using the Folch method and separated using TLC, and FAMEs were then derived from sphingomyelin, phosphatidylcholine, and cholesteryl ester fractions. Unlike Suh and Agbaga, Serrano et al. used soft ionization and performed peak acquisition using a form of tandem mass spectrometry, MS^E^, which obtains accurate mass for precursor and product ions using low (4 eV) and high-energy (25–20 eV) collision acquisition functions. They validated their experiments by recovering VLC-PUFA peaks after internal standard (C27:0) and VLC-SFA (C28:0–C30:0) fortification of *Sparus aurata* eye samples and re-studying mass spectra [85]. Through their previous work, in 2021, Serrano et al. identified VLC-PUFAs up to 44 carbons in chain length from as little as 2.5 × 10^−3^ pmol/mg lipid [35].

### 8.2. LC-MS/MS

A second way to analyze VLC-PUFAS is through liquid chromatography-tandem mass spectrometry (LC-MS/MS). VLC-PUFAs in glycerophospholipids can also be analyzed through normal-phase HLPC-MS with mass spectrometry conducted in negative electrospray ionization mode and multiple reaction monitoring (MRM) mode using precursor and fragment ion *m*/*z* pairs for specific glycerophospholipids [86]. Berdeaux et al. have also quantified underivatized and unpurified VLC-PUFA-PCs using HPLC and liquid chromatography-tandem mass spectrometry with electrospray ionization (LC-ESI-MS/MS) and acquired peaks using collision-induced dissociation (CID) in the negative mode [80]. VLC-PUFAs from porcine photoreceptor outer segments have been quantified using flame ionization and ultra-high-performance tandem liquid chromatography [38]. VLC-PUFAs have also been analyzed in the eyes of a zebrafish model of Zellweger syndrome. In this study, Takashima et al. identified VLC-PUFAs up to 44 carbons in chain length and with up to 13 double bonds in their fatty acyl chains [5]. They used a combination of normal phase ultraperformance liquid chromatography-mass spectrometry (UPLC-MS) in negative-ion electrospray ionization mode. In addition, time-of-flight MS was used to obtain mass spectra for ions between 100–1000 *m*/*z* [5]. Moreover, Hamano et al. analyzed phospholipids containing VLC-FAs in CD73-sorted mouse retinal cells using liquid chromatography and tandem mass spectrometry (LC-MS/MS) with selected reaction monitoring (SRM) [69]. Similarly, N32 and N34 elovanoids have been analyzed in human RPE cells using LC-MS/MS by comparing peak retention times and mass spectra to chemically synthesized elovanoid compounds [87].

VLC-FAs have also been analyzed using Orbitrap mass spectrometry and LC-MS/MS [88] because Busik et al. [88] posited that the current GC-MS methods lacked the sensitivity, high resolution, and robustness of currently available analytical technology. They recently developed a way to analyze more complex lipid groups containing VLC-PUFAs using direct infusion of underivatized fatty acids, liquid chromatography with triple quadrupole Orbitrap mass spectrometry (MS/MS), untargeted “ion mapping” MS/MS, and downstream data-mining. The downside of this method is that large data sets are generated, which requires extensive post-acquisition data processing [88].

Since commercial VLC-PUFA standards are limited in availability, VLC-PUFAs can be quantified using a combination of bovine retina (Figure 5) and odd-chain length VLC-SFAs as standards to calibrate VLC-PUFA concentrations when they are low in abundance in natural systems. As more VLC-PUFAs are identified and studied, these new chemical species and their corresponding mass spectra will be incorporated into the National Institute of Standards and Technology (NIST) lipid ontology databases.

### 8.3. Other Analytical Methods

In 2007, McMahon et al. analyzed underivatized phosphatidylcholine-containing VLC-PUFAs in the retinas of their STGD3 knock-in mice by electrospray ionization and tandem mass spectrometry using a Q-TOF Global Ultima mass spectrometer (ESI-MS/MS). They also employed LiCl salt to get better fragmentation patterns and structural understanding of their VLC-PUFA-containing PCs [79]. Their analyses demonstrated that a heterozygous STGD3 mutation causes selective deficiency of C32-C36 acyl PCs in the mouse retina. Similarly, other groups, including Berdeaux et al. [4], have used 4,4-dimethyloxazoline (DMOX) to make nitrogen-containing derivatives of VLC-PUFAs to analyze their structure and double-bond positions better [4]. These studies confirmed Aveldaño’s findings by demonstrating that the C32 and C34 retinal VLC-PUFAs in human, mouse, and bovine retinas are mainly *n*-3 or *n*-6 tetraenes, pentaenes, and hexaenes.

Another soft ionization technique recently used to characterize VLC-PUFAs qualitatively is matrix-assisted laser desorption/ionization imaging mass spectrometry (MALDI-IMS). MALDI-IMS uses special laser-absorbing matrices to minimize ion fragmentation in frozen sections. Berry et al. [40] used MALDI-IMS to determine the localization of different lipids, including VLC-PUFA-containing phosphatidylcholines within the different retinal layers and optic nerve. Frozen retinal sections are embedded in the MALDI matrix. Mass spectra are then captured by rastering the MALDI laser across areas of interest in the tissue. Using MALDI-IMS, Berry et al. found evidence in support of VLC-PUFA localization to photoreceptor outer segments. In 2020, Vidal et al. used MALDI-IMS to trace the retinal and sub-retinal localizations of dietary fatty acid supplements in the retina. They found a tissue-dependent differential uptake and incorporation of omega-3 fatty acids in triacylglycerols compared to phospholipid formulations [81]. They secondarily used gas chromatography with heated electrospray ionization (H-ESI) and Orbitrap FUSION™ mass spectrometry to characterize triacylglycerols and phospholipids isolated from serum, RPE, and retinal tissue.

Ultimately, the decision on what method to use depends on access to available technologies, sample state (e.g., homogenates vs. tissue sections), and the desired readouts from the assay. While LC-MS/MS in combination with soft-ionization techniques allows for more robust detection of the different chemical forms of VLC-PUFAs, GC-MS combined with hard ionization allows for the quantification of VLC-PUFAs in minute quantities with predetermined structures and fragmentation patterns. A combination of analysis, derivatization, and detection methods, with validated chemical standards, may be used to capture the diversity of VLC-PUFAs in their various chemical forms and degrees of unsaturation.

## 9. VLC-PUFAs in Relation to Cholesteryl Esters and Acylcarnitines in Eye Health

Metabolic changes that occur with aging and disease can also disrupt the homeostatic balance of lipids in sensitive tissue such as the retina. AMD is associated with abnormal retinal morphology and sub-RPE lipid deposits [89]. These abnormal lipid deposits can also contain minerals and protein and block nutrient and waste transfer between the photoreceptors and RPE [90]. The photoreceptors in these nutrient-deficient conditions cannot sustain themselves, leading to the retinal degeneration observed in AMD and STGD3 patients. Besides phospholipid-containing VLC-PUFAs, lipid groups like cholesteryl esters, cholesterols (e.g., 7-ketocholesterol, 7-KC), and acylcarnitines have been implicated in eye health and disease.

Cholesteryl esters regulate lipid metabolism by being involved in cholesterol storage and transportation. Cholesteryl esters are also elevated in specific brain tissue of patients with neurodegenerative diseases such as Huntington’s [91]. In humans, cholesterol accumulates in Bruch’s membrane with age and can form the retinal lesions associated with AMD [92]. Neutral lipid accumulation in the retina can be visualized with Oil Red O staining in different ocular tissue. Cholesterol can be biosynthesized in the retina by glial cells or taken up through the RPE and blood-retina barrier via lipoprotein transporters [93]. Increased DHA intake is associated with lower levels of inflammation induced by cholesterol-rich diets in the eye and brain [94]. Although VLC-PUFAs can exist in cholesteryl esters [95], the relationship between VLC-PUFA uptake and cholesterol accumulation remains to be investigated.

Elevated acylcarnitine levels are associated with dysfunctional fatty acid oxidation and metabolic disorders such as Type II diabetes [96,97,98]. Acylcarnitines play an important role in lipid metabolism via lipid transport to the mitochondria for β-oxidation [99]. They come in different lengths, degrees of saturation, and branching; these characteristics influence their function, health, and disease in different tissue [100]. Although it is unclear if acylcarnitines are synthesized endogenously in the retina, their dysregulation is associated with many vision disorders. For instance, patients with neovascular or wet AMD have reduced plasma levels of short-chain (C2) acylcarnitines and increased long-chain acylcarnitine levels [101]. The dysregulation of acylcarnitine is also implicated in other ocular diseases, such as diabetic retinopathy, retinopathy of prematurity, primary open-angle glaucoma, and dry eye syndrome, among others [102].

Furthermore, VLC-FA-containing -acylcarnitines are acylcarnitines with very-long-chain saturated, monounsaturated, and polyunsaturated fatty acyl moieties. Due to their chain length, these lipids are primarily metabolized by peroxisomes instead of mitochondrial β-oxidation like their shorter chain counterparts [100,103]. In patients with type II diabetes and some peroxisomal disorders, specific classes of very long-chain acylcarnitines are dysregulated in blood and plasma lipid extracts [100]. There is also an apparent gender- and age-dependent accumulation of long-chain- and very-long-chain- fatty acid-containing acylcarnitines in certain tissue [104]. Also, ELOVL1 is known to produce very long-chain branched-chain fatty acids typically incorporated in ceramides in the meibomian glands and liver [105]. However, their physiological role in these organs and their presence in the eye still need to be determined. The complexity of VLC-PUFAs in different systems would require a high-resolution untargeted lipidomic approach to characterize.

Since LC-PUFAs have some benefits in slowing down the progression of retinal disorders, it would be interesting to determine whether dietary supplementation of VLC-PUFAs would also improve vision in disease models. More research needs to be conducted on VLC-PUFA transport pathways, metabolism, and tissue incorporation to determine which delivery method (e.g., oral gavage, intravitreal injections, or eye drops) and chemical form (such as cholesteryl ester, triacylglycerol, phospholipid, etc.) would be most efficiently incorporated in targeted organs.

## 10. Conclusions

The translational value of VLC-PUFAs is their inherent potential for treatment against retinopathies and other conditions where the loss of VLC-PUFAs is implicated in disease etiology. There is a gap in non-invasive therapies against AMD. The addition of omega-3 fatty acids in the AREDS2 Study did little to improve the visual outcomes of patients with AMD, compared to formulations without the fatty acid supplement [106]. This could be due to poor uptake, transportation, or conversion of the omega-3 fatty acids to VLC-PUFAs in the retina. With more research and dose and delivery optimization, the positive outcomes observed with VLC-PUFA supplementation and Elovl4 transgenes in cell culture and animal studies can translate to positive outcomes in human disease conditions.

Although much remains unknown about the physiological role of VLC-PUFAs in human retinas, the development of new technologies and progress in integrative, science-driven approaches to studying lipid dysregulation yield promise for elucidating the role of VLC-PUFAs in eye health and their potential therapeutic uses. Moreover, much remains unknown about the role of desaturases, peroxisomes, transporters, and other players involved in VLC-PUFA metabolism. It would be interesting to investigate if VLC-PUFAs play a role in other inherited retinal dystrophies like macular telangiectasia and other forms of Stargardt Disease.

An integrative approach is needed to improve our current understanding of the role of VLC-PUFAs in eye health, one which combines structural and functional studies and reflects clinical data. ELOVL4 pathologies such as STGD3, ichthyosis, and spinocerebellar ataxia-34 highlight the importance of VLC-PUFAs in normal human development and function. Retinal degenerations such as AMD are multi-faceted, complex and involve many cell types and extrinsic factors. Thus, it is important to study the contributions of each cell type and factor to VLC-PUFA pathologies. In addition, it is important to know how VLC-PUFAs, in various chemical forms, derivatives, concentrations, and tissue distributions, can be used as biomarkers for overall health.

## Figures and Tables

**Figure 1 nutrients-15-03096-f001:**
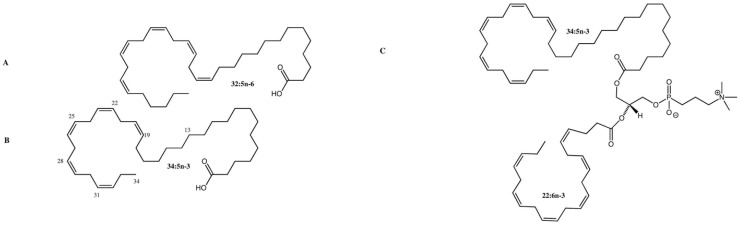
Structure of C32 and C34 VLC-PUFAs in the human retina. (**A**) C32:5 n-6 VLC-PUFA; (**B**) C34:5 n-3 VLC-PUFA; (**C**) Phosphatidylcholine containing C34:5 n-3 VLC-PUFA (sn-1) and DHA (sn-2). Methylene-interrupted cis double bonds are located on the distal ends of VLC-PUFAs, while the proximal ends are reminiscent of saturated fatty acids. Images adapted from Gorusupudi et al. [8] with authors’ permission.

**Figure 2 nutrients-15-03096-f002:**
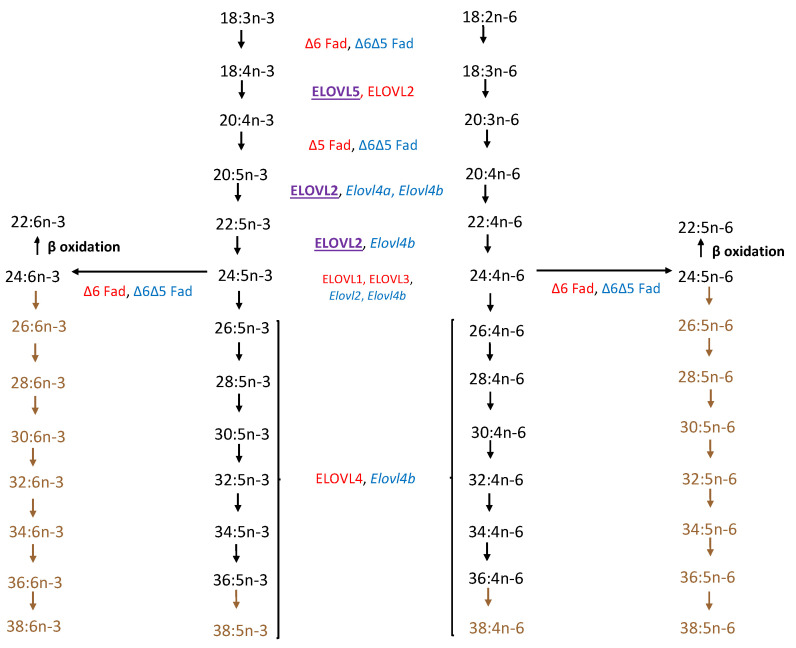
Comparison of biosynthetic pathway of VLC-PUFAs from linoleic (18:2 n-6) and α-linolenic (18:3 n-3) acid precursors of zebrafish and mammalian vertebrates (adapted from Monroig et al. [9] and Hopiavuori et al. [10]). Color Key: Red: Mammalian (human and murine); Blue: Zebrafish; Purple: Both; Brown: Pathway presumed but not confirmed in zebrafish.

**Figure 3 nutrients-15-03096-f003:**
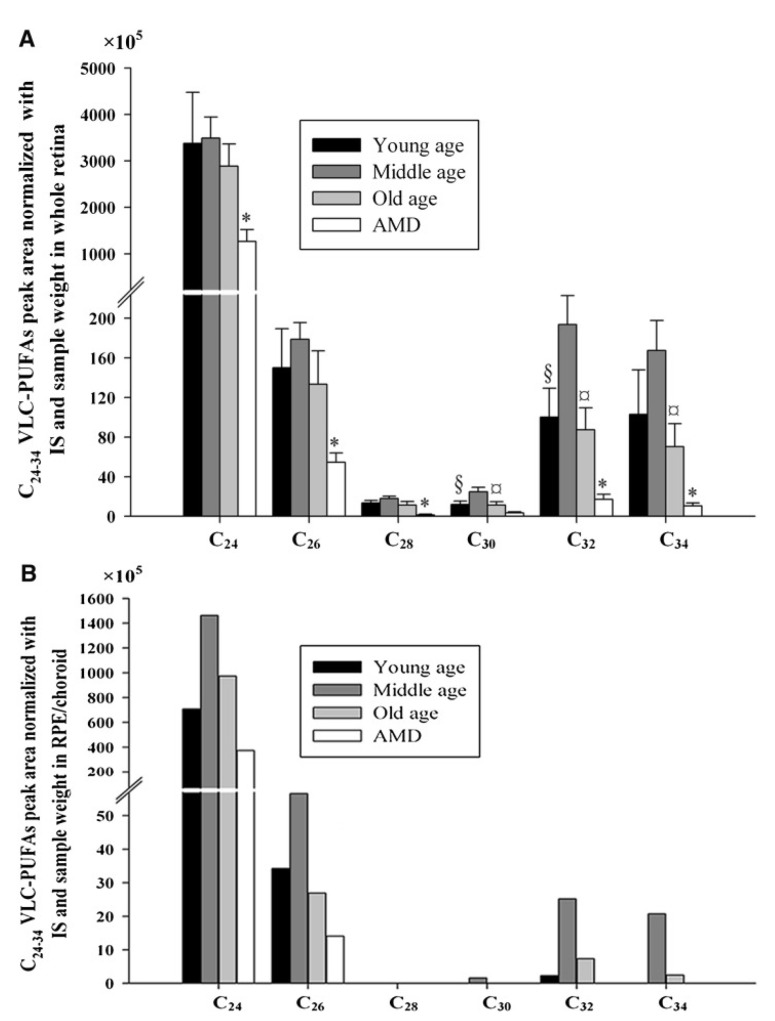
C24-C34 VLC-PUFA comparison in lipids extracted from whole retinas (**A**) and RPE/choroidal tissue (**B**) obtained for normal young, middle-aged, and old individuals and age-matched patients with AMD. This image shows that patients with AMD had significantly lower retinal C24, C26, C28, C32, and C34 VLC-PUFAs than age-matched controls. Moreover, VLC-PUFAs were also reduced in the RPE/choroid of patients with AMD. §, Significant differences (*p* < 0.05) between young and middle age group; ¤, means significant differences (*p* < 0.05) between middle and old age group; *, significant differences (*p* < 0.05) between old and age-matched AMD group. In (**B**), there are no statistics provided, because each group’s samples had to be pooled. IS: internal standard. Image obtained from Liu et al. [16] with authors’ permission.

**Figure 4 nutrients-15-03096-f004:**
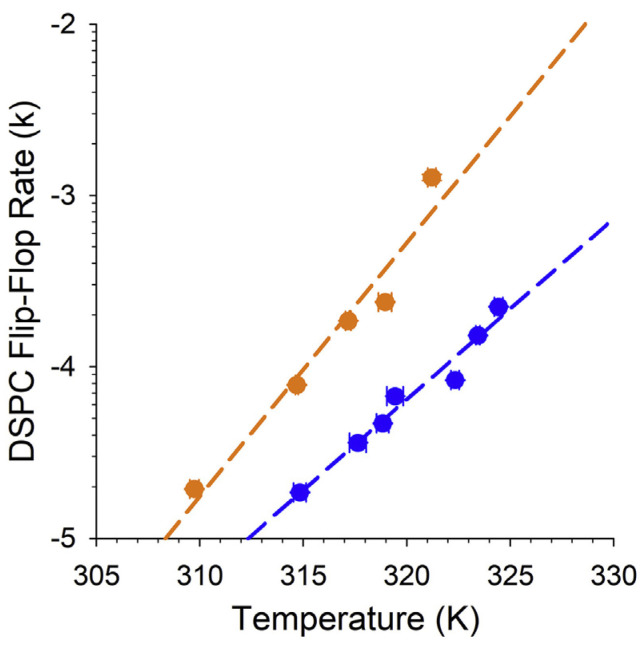
VLC-PUFAs increase lipid flip-flop in synthetic bilayers. Comparison of the changes in deuterium-labeled distearoylphosphatidylcholine (DSPC) flip-flop rate (k) with increasing temperature in synthetic DSPC lipid bilayers supplemented with 0.1% C32:6 n-3 VLC-PUFA (yellow) and non-supplemented (blue) DSPC bilayer. Image obtained from Cheng et al. [52] with permission.

**Figure 5 nutrients-15-03096-f005:**
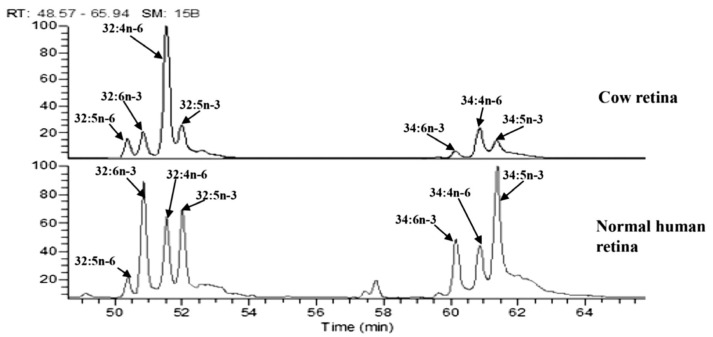
Retention times from VLC-PUFAs isolated from bovine retinas are matched to retention times in other retinal tissue. Analysis of C32 and C34 VLC-PUFAs in the human retina using bovine retina to match VLC-PUFA peak retention times. Image adapted from Gorusupudi et al. [83] with authors’ permission.

## Data Availability

The data presented in this review are openly available in the peer-reviewed journals listed in the “References” section.

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
