# Peer review of "Understanding the Roles of Very-Long-Chain Polyunsaturated Fatty Acids (VLC-PUFAs) in Eye Health"

_nutrients, 2023, doi:10.3390/nu15143096_

Round 1

Reviewer 1 Report

The review nicely summarized the current understanding and progress of very-long-chain polyunsaturated fatty acids (VLC-PUFAs) in the field of retina and the potential for targeted therapies, including AMD, RP and DR etc. Several minor comments as below:

1. In section 8, the authors covered the previous and emerging VLC-PUFA analysis tools. How easy/difficult to perform these analyses? And what is the pros and cons of each technology?

2. What are, in general, the clinical applications or translational value of VLC-PUFAs?

Author Response

Reviewer 1

  • Comment 1. In section 8, the authors covered the previous and emerging VLC-PUFA analysis tools. How easy/difficult to perform these analyses? And what is the pros and cons of each technology?

Author response: We appreciate the reviewer’s comments and acknowledge the benefits of a comparative study of the methods employed in VLC-PUFA analysis. We also recognize that the current methods of VLC-PUFA analysis were refined and built on existing methods. A detailed comparison and evaluation of the ease and difficulty of each technique would require intimate knowledge and experience with the different sample preparation and analysis software and techniques.

However, from the literature, we can compare the methods in terms of critical readouts like maximum VLC-PUFA length detected, the limit of detection, robustness, and ability to detect VLC-PUFAs of differing chemical species, structure, and degrees of unsaturation. Moreover, the methods used depend on the application, e.g., detecting VLC-PUFAs in tissue sections vs. homogenates.  

We have chronologically outlined the most recent methods, their capabilities, and the benefits and applications of soft vs. hard ionization techniques. However, it is ultimately up to the reader to decide what method to choose based on ease and affordability, start-up and running costs, etc., for their specific applications.

To explain this, we added an extra paragraph to the section, highlighted on page 12, lines 503-511.

  • Comment 2. What are, in general, the clinical applications or translational value of VLC-PUFAs?

Author response: The general translational value of VLC-PUFAs is in their potential for treatment against retinopathies and other conditions where the loss of VLC-PUFAs is implicated in disease etiology. There is a gap in non-invasive therapies against age-related macular degeneration (AMD). The addition of omega-3 fatty acids in the AREDS2 Study did little to improve the visual outcomes of patients with AMD, compared to formulations without the fatty acid supplement, and this could be due to a poor uptake, transportation, or a poor conversion rate VLC-PUFAs in the retina. Our lab's studies seem promising in that the dietary supplementation of C32: n-6 VLC-PUFA is retinally bioavailable and improves ERG responses in fed wild-type mice compared to controls.  With more research and dose and delivery optimization, the positive outcomes observed in cell culture and animal studies can potentially translate to positive outcomes in human disease conditions.

We have re-emphasized the translational value of VLC-PUFAs in the article's conclusion and highlighted these changes on page 13, lines 565-574.

Reviewer 2 Report

Nwagbo and Bernstein present a comprehensive review on the biochemistry, physiology and pathophysiology of very long chain fatty acids in the retina. They also include a useful section on technologies to analyze these fatty acids. The paper is well researched and discussed. The citation of the literature appears appropriate. I do have a few minor comments.

1) In the caption to Fig. 1C, it is stated that the PC molecule contains C34:5n-3 at the sn-2 position, and DHA at the sn-1 position. The figure shows the contrary. According to lines 142-143, it is the caption that appears to be incorrect.   

2) The bibliographical references included in the captions to Figs. 1, 2 and 4 are not in the appropriate format. The same problem occurs in lines 415 and 451. Include the citation number in parentheses, not the publication year.

3) Line 70.  The authors probably meant “docosanoids”, as the eicosanoids derive from C20 fatty acids.

4) Fig. 3 is too small. Please enlarge it.

5) Line 145. Use of the term “triacylglycerol” is vastly preferable over “triacylglyceride”.

6) Line 157. Please qualify this sentence by indicating that “peroxisomes are responsible for very-long fatty acid β-oxidaton and degradation.” The major organelle for fatty acid β-oxidation and degradation is the mitochondria.

Author Response

Reviewer 2

  • Comment 1) In the caption to Fig. 1C, it is stated that the PC molecule contains C34:5n-3 at the sn-2 position, and DHA at the sn-1 position. The figure shows the contrary. According to lines 142-143, it is the caption that appears to be incorrect.

Author response: Thank you for your corrections. We have updated and highlighted the caption for Fig 1C, lines 52 and 53 on pg. 2.

  • Comment 2) The bibliographical references included in the captions to Figs. 1, 2 and 4 are not in the appropriate format. The same problem occurs in lines 415 and 451. Include the citation number in parentheses, not the publication year.

Author response: Thank you for this observation. We have made the changes and highlighted the text on these figures in lines 415 and 451 in the article's main body. The updates to these citations can be found on lines 54 and 58-59, page 2; line 229, page 6; line 422, page 10; and line 458, page 11. Additionally, we extended these changes to Fig. 3, page 4, lines 121-122; Fig. 5, page 11, line 474, line 127, page 4; and lines 483 and 491, page 12.

  • Comment 3) Line 70. The authors probably meant “docosanoids”, as the eicosanoids derive from C20 fatty acids.

Author response: Thank you for this correction. We have changed and highlighted the term on line 70 of page 3.

  • Comment 4) Fig. 3 is too small. Please enlarge it.

Author response: Thank you. We have enlarged Fig.3 (pg. 4) accordingly.

  • Comment 5) Line 145. Use of the term “triacylglycerol” is vastly preferable over “triacylglyceride”.

Author response: Thank you for this observation. We have highlighted and changed the word “triacylglyceride”, now found on line 149, page 5. We have also changed the words “triacylglycerides” and “triglycerides,” previously located on lines 498 and 501, to “triacylglycerols.”  These new words are highlighted in the text on page 12.

  • Comment 6) Line 157. Please qualify this sentence by indicating that “peroxisomes are responsible for very-long fatty acid β-oxidaton and degradation.” The major organelle for fatty acid β-oxidation and degradation is the mitochondria.

Author response: Thank you for your observation. We have made these changes and highlighted the text on lines 161-162 on page 5.